



# Modelling concentration heterogeneities in streets using the street-network model MUNICH

Thibaud Sarica[1], Alice Maison[1,2], Yelva Roustan[1], Matthias Ketzel[3], Steen Solvang Jensen[3], Youngseob Kim[1], Christophe Chaillou[4], and Karine Sartelet[1]

[1]CEREA, École des Ponts ParisTech, EDF R&D, IPSL, Marne-la-Vallée, 77455, France
[2]Université Paris-Saclay, INRAE, AgroParisTech, UMR EcoSys, 91120 Palaiseau, France
[3]Department of Environmental Science, Aarhus University, Roskilde, Denmark
[4]Aramco Fuel Research Center, Aramco Overseas Company, Rueil-Malmaison, 92500, France

**Correspondence:** Thibaud Sarica (thibaud.sarica@enpc.fr), Karine Sartelet (karine.sartelet@enpc.fr)

**Abstract.** Populations in urban areas are exposed to high local concentrations of pollutants, such as nitrogen dioxide and particulate matter, because of unfavorable dispersion conditions and the proximity to traffic. To simulate these concentrations over cities, models like the street-network model MUNICH (Model of Urban Network of Intersecting Canyons and Highways) rely on parameterizations to represent the air flow and the concentrations of pollutants in streets. In the current version MUNICH v2.0, concentrations are assumed to be homogeneous in each street segment. A new version of MUNICH where the street volume is discretized is developed to represent the street gradients and better estimate people exposure. Three vertical levels are defined in each street segment. A horizontal discretization is also introduced under specific conditions by considering two zones with a parameterization taken from the Operational Street Pollution Model (OSPM). Simulations are performed over two districts of Copenhagen, Denmark, and one district of Greater Paris, France. Results show an improvement of the comparison to observations with higher concentrations at the bottom of the street, closer to traffic, of pollutants emitted by traffic ($NO_x$, black carbon, organic matter). Finally, a sensitivity analysis to the influence of the street network highlights the importance to use the model MUNICH with a network rather than with a single street.

## 1 Introduction

Pollution is estimated to be responsible for approximately 9 million premature deaths in 2015 (Landrigan et al., 2018). This figure remains valid in 2019 despite an improvement of the types of pollution associated with extreme poverty (e.g., household air pollution and water pollution) (Fuller et al., 2022). This is partly due to an increase in the number of premature deaths attributable to ambient air pollution. The consequences of air pollution are particularly substantial in urban areas, where individuals are exposed to local high concentrations of air pollutants due to unfavorable dispersion conditions and proximity to traffic. As more than half of the world's population already lives in urban areas, rising to 68 % by 2050 (United Nations, 2019), it is crucial to estimate as accurately as possible the exposure of population to atmospheric pollutant concentrations in urban areas. For many years, various modelling approaches have been developed to contribute to the understanding of the phenomena



that drive the concentrations of pollutants in the atmosphere and to provide decision support tools (Collett and Oduyemi, 1997; Vardoulakis et al., 2003; El-Harbawi, 2013; Conti et al., 2017; Khan and Quamrul, 2021).

Regional-scale chemistry-transport models, such as Polair3D (Mallet et al., 2007; Sartelet et al., 2018), CHIMERE (Menut
et al., 2021; Falasca and Curci, 2018), CMAQ (Wong et al., 2012; de la Paz et al., 2015), represent the urban background concentrations by solving the chemistry-transport equation for spatial resolutions down to $1\,\mathrm{km}^2$. However, they can not represent street concentrations, which are often higher than background concentrations for pollutants such as $NO_2$ and particles (Lugon et al., 2020). To represent these concentrations, local-scale models are thus developed with different approaches of variable complexity and computational cost. Models based on computational fluid dynamics (CFD), such as code_saturne (Archambeau
et al., 2004; Gao et al., 2018), OpenFoam (Lin et al., 2022) and PALM (Wolf et al., 2020), are able to represent the dispersion of pollutants and the physicochemical processes taking place in urban districts and streets with a fine spatial resolution by solving the Navier-Stokes equations and mass conservation equations for pollutants. However, they suffer from high computational cost, as they use fine meshes to describe the morphology of buildings and streets. Other models use approaches that are less accurate than CFD but run faster. They are typically based on a Gaussian or an Eulerian approach (Vardoulakis et al., 2003;
Liang et al., 2023). Among these, can be mentioned ADMS-Urban (McHugh et al., 1997; Hood et al., 2021), SIRANE (Soulhac et al., 2011, 2023) and AERMOD (Cimorelli et al., 2004; Rood, 2014). They either consider each street independently of the others with exchanges between the street and the background concentrations above the street (Berkowicz, 2000a), or a street network with incoming/outcoming flows between streets at intersections (Soulhac et al., 2009; Kim et al., 2022). The Operational Street Pollution Model (OSPM) couples a Gaussian-plume model for traffic emissions and a box model for the
recirculation in the street (Berkowicz, 2000a). It is thus able to represent concentration heterogeneities in the street, but cannot include complex chemistry. The Model of Urban Network of Intersection Canyons and Highways (MUNICH) uses solely an Eulerian box-model approach (Lugon et al., 2020). As MUNICH is coupled with the SSH-aerosol model (Sartelet et al., 2020), the formation and aging of primary and secondary gas and particles in streets are represented. In the current version of MUNICH (v2.0) (Kim et al., 2022), concentrations are considered homogeneous in each street segment. However, as shown
by several on-site and modelling studies, the concentrations are very heterogeneous in streets with traffic emissions (Xie et al., 2003; Vardoulakis et al., 2011; Lateb et al., 2016; Sanchez et al., 2016; Amato et al., 2019; Lin et al., 2022). They are higher near the ground than at the top of the street, especially for primary pollutants.

A "heterogeneous" version of MUNICH is developed in this study aiming to represent the concentration heterogeneities in the street, while keeping the Eulerian approach of MUNICH to retain the ability to accurately model chemistry and aerosol
dynamics. The street volume is discretized vertically in three subvolumes. Traffic emissions are not instantaneously diluted in the whole street volume as in MUNICH v2.0, but only in the first subvolume, i.e. the one closest to the ground. To represent the street horizontal heterogeneities, a recirculation zone of the shape of a trapeze is defined, based on a parameterization of OSPM. It depends on the meteorological conditions and the street morphology, and it is applied under specific conditions in MUNICH that are described later.

A description of the differences between the homogeneous version of MUNICH (v2.0) and the new heterogeneous version is presented in Section 2. The applications to two street networks in Copenhagen, Denmark, with comparisons to observations





of NO$_2$ and CO, and to concentrations simulated by OSPM are discussed in Section 3. MUNICH is applied in Section 4 to the street network near Paris, France, used to validate MUNICH v2.0 (Kim et al., 2022), and the impacts of the heterogeneous version on concentrations of NO$_2$ and particles are studied. To estimate the impact of the modelling hypothesis on the transport

of pollutants between streets in the new version of MUNICH, a sensitivity analysis to the presence of a street network is performed in Section 5.

## 2   Model description

The homogeneous version of MUNICH (v2.0) and the new heterogeneous version (see Fig. 1) are briefly described in this section, focusing on the differences between the two versions. In the following, the homogeneous and heterogeneous versions

of MUNICH are referred to as MUNICH-homo and MUNICH-hete respectively. For a complete description of MUNICH, please refer to Kim et al. (2018, 2022) and Lugon et al. (2020, 2021a).

In order to solve the evolution equation of the street concentrations $C_{\mathrm{str}}$, a first-order operator splitting between transport and chemistry is performed:

$$\frac{\mathrm{d}C_{\mathrm{str}}}{\mathrm{d}t} = \left.\frac{\mathrm{d}C_{\mathrm{str}}}{\mathrm{d}t}\right|_{\mathrm{tr}} + \left.\frac{\mathrm{d}C_{\mathrm{str}}}{\mathrm{d}t}\right|_{\mathrm{ch}} \tag{1}$$

At each time step, the transport component is computed first. Chemical transformations are then applied to the resulting concentrations. As deposition and resuspension processes have minor effects compared to transport and chemistry (Lugon et al., 2021b; Kim et al., 2022), they are omitted in the rest of this study.

### 2.1   Homogeneous approach

Using a box-model approach, the concentrations are assumed to be homogeneous in the whole street volume and the effect of

the processes on the concentrations are represented by the equation:

$$\left.\frac{\mathrm{d}C_{\mathrm{str}}}{\mathrm{d}t}\right|_{\mathrm{tr}} = \frac{1}{V}\left(Q_{\mathrm{em}} + Q_{\mathrm{inflow}} + Q_{\mathrm{outflow}} + Q_{\mathrm{vert}}\right) \tag{2}$$

with $V$ the volume of the rectangular cuboid street, $Q_{\mathrm{em}}$ the traffic emission flux, $Q_{\mathrm{inflow}}$ the flux entering the street via the upwind intersection, $Q_{\mathrm{outflow}}$ the flux leaving the street via the downwind intersection and $Q_{\mathrm{vert}}$ the vertical turbulent flux between the background and the street.

The street volume is defined as $V = HWL$, with $H$ the mean building height in the street, $W$ the mean street width and $L$ its length. MUNICH considers that buildings on each side of the street are continuous, thus not representing inflow/outflow that could be induced by gaps in the buildings. The inflow term $Q_{\mathrm{inflow}}$ is obtained from the computation of the fluxes at the upwind intersection (Kim et al., 2018; Soulhac et al., 2009) (see Sec. 5 for a brief description). The outflow term $Q_{\mathrm{outflow}}$ is expressed as :

$$Q_{\mathrm{outflow}} = HW u_{\mathrm{str}} C_{\mathrm{str}} \tag{3}$$





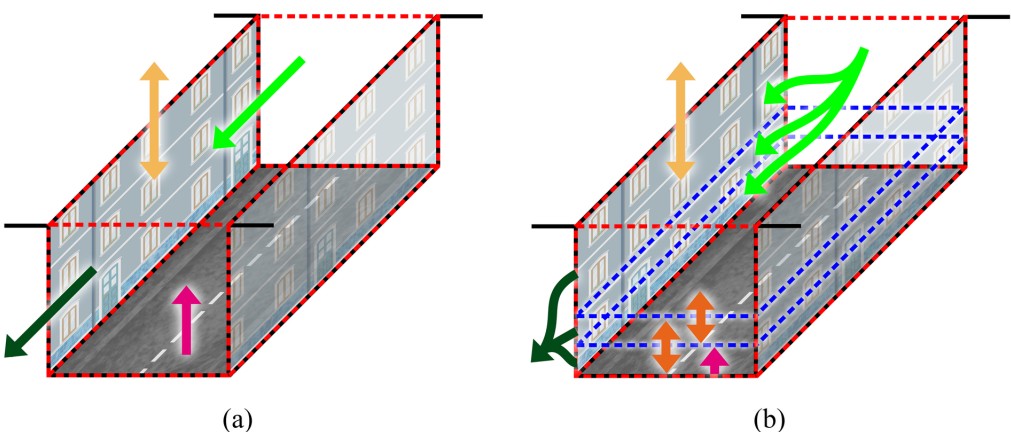

**Figure 1.** Representation of the processes in the homogeneous version (a) and the heterogeneous version (b) of MUNICH. The red dotted lines represent the street volume. In (a), the street canopy is represented by a single volume, whereas in (b), it is divided into 3 subvolumes delimited by the blue dotted lines. The rose arrows represent the traffic emissions, the light green arrows the fluxes entering the street via the upwind intersection and the dark green arrows the fluxes leaving the street via the downwind intersection. The yellow arrows symbolize the vertical turbulent exchanges with the background and, in (b), the orange arrows the vertical exchanges among the subvolumes.

with $u_{\mathrm{str}}$ the mean horizontal wind speed in the street.

The vertical turbulent flux $Q_{\mathrm{vert}}$ between the street and the overlying atmosphere is:

$$Q_{\mathrm{vert}} = q_{\mathrm{vert}} W L \frac{(C_{\mathrm{str}} - C_{\mathrm{bkgd}})}{H} \tag{4}$$

with $q_{\mathrm{vert}}$ the vertical transfer coefficient and $C_{\mathrm{bkgd}}$ the background concentration.

The vertical transfer coefficient and the horizontal wind speed are the key parameters representing the dispersion of concentrations. As their formulation differs between the homogeneous and the heterogeneous versions of MUNICH, they are now detailed.

### 2.1.1 Vertical transfer coefficient for turbulent flux

Three parameterizations are implemented in MUNICH to determine the vertical transfer coefficient, $q_{\mathrm{vert}}$, between the street and the overlying atmosphere (Maison et al., 2022). However, currently only the parameterization adapted from Wang (2014) is designed to provide vertical profiles for both wind speed and mixing length within the street. We therefore limit our analysis to the latter.

In MUNICH-homo, the vertical transfer coefficient at the roof level is expressed as:

$$q_{\mathrm{vert}} = \sigma_{\mathrm{w}} l_{\mathrm{m}}(z = H) \quad \text{with} \quad l_{\mathrm{m}}(z) = \frac{\kappa z \, l_{\mathrm{c}}}{l_{\mathrm{c}} + \kappa z} \tag{5}$$





where $\sigma_{\mathrm{w}}$ is the standard deviation of the vertical wind velocity at roof level, $l_{\mathrm{m}}$ is the mixing length defined as a harmonic mean between two length scales (Coceal and Belcher, 2004) i) $\kappa z$, with the Von Kármán constant ($\kappa = 0.42$) and ii) $l_{\mathrm{c}}$ a characteristic length of the street chosen equals to $0.5W$ (Maison et al., 2022).

### 2.1.2   Mean horizontal wind speed in the street

As for the calculation of the vertical transfer coefficient, three parameterizations are proposed in MUNICH to determine the
mean horizontal wind speed in the street (Maison et al., 2022). In the *Wang* parameterization, the mean horizontal wind speed in the street $u_{\mathrm{str}}$ is equal to:

$$u_{\mathrm{str}} = \frac{1}{H - z_{0s}} \int\limits_{z_{0s}}^{H} u_Y(z)\,\mathrm{d}z = \frac{u_H |cos(\varphi)|}{(H - z_{0s})} \int\limits_{z_{0s}}^{H} [J_1 I_0(g(z)) + J_2 K_0(g(z))]\,\mathrm{d}z \tag{6}$$

with $u_H |cos(\varphi)|$ the wind speed at roof level in the direction of the street and $z_{0s}$ the wall and ground roughness length in the street (fixed to 0.01 m). $I_0$ and $K_0$ are the first and second kind modified Bessel function of order 0. $J_1$ and $J_2$ are integration
coefficients equal to:

$$J_1 = \frac{1}{I_0(g(H)) - I_0(g(z_{0s}))K_0(g(H))/K_0(g(z_{0s}))} \text{ and } J_2 = -\frac{J_1 I_0(g(z_{0s}))}{K_0(g(z_{0s}))} \tag{7}$$

The function $g(z)$ is calculated as:

$$g(z) = 2\sqrt{C_B a_r \frac{z}{l_{\mathrm{m}}(H)}} \tag{8}$$

with $a_r = H/W$ the aspect ratio of the street and $C_B$ a coefficient dependent on the wind angle with the street and the aspect
ratio (Maison et al., 2022).

### 2.2   Heterogeneous approach

In MUNICH-hete, the street is divided into 3 vertical levels to limit the artificial dilution of the traffic emissions and the concentrations in the whole street volume (see Fig. 1(b) and Fig. 2). Levels are ordered from the ground to the top of the street. The first level (i=1) contains the traffic emissions. The thickness $h_1$ is taken as 2 m, which correspond to a zone where the
traffic producing turbulence mixes and dilutes traffic emissions (Solazzo et al., 2008). The second level (i=2) thickness $h_2$ is also of 2 m. It acts as a buffer zone between the first level where traffic emissions are and the third level where exchanges with the background take place. Starting at 4 m, the third level (i=3) goes to the roof level ($h_3 = (H - 4)$ m). The minimum street height considered in the model is set at $6\,m$. The three levels of the heterogeneous version are referred to as munich-hete-l1, munich-hete-l2 and munich-hete-l3 respectively. Each level i is thus associated to a specific volume, $V_i$, and the evolution
equations may be written as:

$$\left.\frac{\mathrm{d}C_{\mathrm{str}}^i}{\mathrm{d}t}\right|_{\mathrm{tr}} = \frac{1}{V_i}\left(Q_{\mathrm{em}}^i + Q_{\mathrm{inflow}}^i + Q_{\mathrm{outflow}}^i + Q_{\mathrm{vert}}^{i,i+1} + Q_{\mathrm{vert}}^{i-1,i}\right) \tag{9}$$





with $Q_{\mathrm{em}}^i$ the traffic emission flux (only in the first level (i=1)), $Q_{\mathrm{inflow}}^i$ the flux entering the level via the upwind intersection, $Q_{\mathrm{outflow}}^i$ the flux leaving the level via the downwind intersection, $Q_{\mathrm{vert}}^{i,i+1}$ the vertical turbulent flux between the levels i and i+1 (for i=3, it exchanges with the background) and $Q_{\mathrm{vert}}^{i-1,i}$ the vertical turbulent flux between the levels i-1 and i (equals to zero if i=1).

In OSPM, the flow developing into a vortex in the street between buildings is represented by a recirculation zone. It occupies the whole street volume for narrow streets, and it has the shape of a trapeze for wider streets (Berkowicz et al., 1997; Berkowicz, 2000a; Ottosen et al., 2015). When the recirculation zone does not occupy the whole street volume, there is a ventilation zone (see Fig. 2) where concentrations of pollutants emitted by traffic are usually lower than in the recirculation zone. In MUNICH-homo, the recirculation zone is not explicited, whatever the street ratio H/W is. In MUNICH-hete, the volume of the recirculation zone is computed as in OSPM, as detailed in Appendix A. For now, concentrations in the ventilation zone are considered homogeneous and equal to the background concentrations, i.e. concentrations above the street. The ventilation zone is thus taken into account when it is not affected by traffic emissions. In practice, this means that the width of the trapeze base can only be equal to or larger than the width $W$ of the street (see App. A). The width $W_i$ of the level i can thus be inferior to the width $W$ of the street, reducing the level volume (see Fig. 2). Appendix A presents the algorithm implemented in MUNICH-hete to consider the volume reduction of the ventilation zone. Further work is needed to differentiate the two zones for cases where the ventilation zone develops further into the street.

To quantify mass transfer through intersection, the fluxes are assumed to be vertically homogeneous and remain determined as proposed by Soulhac et al. (2009), as the shape of intersections may differ from one to another and turbulence is not quantified. The flux $Q_{\mathrm{inflow}}$ entering the street is assumed to be the same for each vertical level. The flux $Q_{\mathrm{outflow}}$ leaving a street is a surface-weighted average of the fluxes leaving each vertical level of the street:

$$Q_{\mathrm{outflow}} = \sum_{i=1}^{i=3} Q_{\mathrm{outflow}}^i = \sum_{i=1}^{i=3} S_i^v u_i C_{\mathrm{str}}^i \tag{10}$$

with $S_i^v$ the vertical surface of the level $i$ as presented in App. A and $u_i$ the mean horizontal wind speed of the level $i$ (see Sec. 2.2.2).

## 2.2.1 Vertical turbulent fluxes

To compute the vertical turbulent transfer $Q_{\mathrm{vert}}^{i,i+1}$ at the interface of the vertical levels i and i+1, Eq. 4 is modified to represent the vertical exchanges between the levels i and i+1:

$$Q_{\mathrm{vert}}^{i,i+1} = q_{\mathrm{vert}}^i W_i L \frac{(C_i - C_{i+1})}{\Delta z_i^{\mathrm{m}}} \tag{11}$$

with $C_i$ and $C_{i+1}$ the concentrations in the levels i and i+1 respectively, $W_i$ the width of the level i that is inferior or equal to the width $W$ of the street and $\Delta z_i^{\mathrm{m}}$ the difference in altitude between the middles of the levels i+1 and i, which are noted $z_1^{\mathrm{m}}$, $z_2^{\mathrm{m}}$ and $z_3^{\mathrm{m}}$ (see Fig. 2). For i=3, i.e. the highest level, $z_{i+1}^{\mathrm{m}}$ is taken as the symmetrical of $z_3^{\mathrm{m}}$ to the roof level, and noted $z_{\mathrm{bkgd}}^{\mathrm{m}}$. It gives an approximation of the volume that effectively exchanges with the third level.



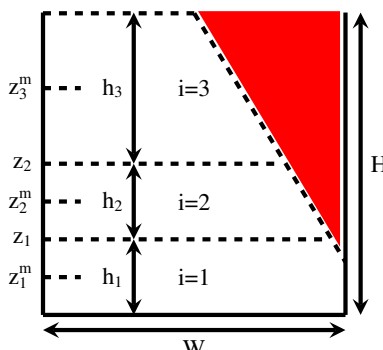

**Figure 2.** Schematic of the discretization in MUNICH-hete. The red triangle represent the ventilation zone that is present under specific conditions, and the white trapeze represents the recirculation zone. $z_i^m$ is the middle height of level i.

Among the three parameterizations implemented in MUNICH to determine the vertical transfer coefficient $q_{\text{vert}}$, only the *Wang* parameterization is adapted to the discretization. It is thanks to its explicit vertical dependency and validity for a wide

range of street-canyon and wind characteristics (Maison et al., 2022). To compute the vertical turbulent transfer $Q_{\text{vert}}^{i,i+1}$, the vertical transfer coefficient is taken at the height of the interface between the two vertical levels considered, and Eq. 5 is now written as:

$$q_{\text{vert}}^i = \sigma_w \kappa z_i \frac{l_c}{l_c + \kappa z_i} \tag{12}$$

with $z_i$ the height of the interface between levels i and i+1.

By combining Eq. 11 and Eq. 12, the vertical turbulent transfer $Q_{i,i+1}$ can be written for each level as:

$$Q_{\text{vert}}^{1,2} = \sigma_w \kappa z_1 \frac{l_c}{l_c + \kappa z_1} W_i L \frac{C_1 - C_2}{z_2^m - z_1^m} \tag{13}$$

$$Q_{\text{vert}}^{2,3} = \sigma_w \kappa z_2 \frac{l_c}{l_c + \kappa z_2} W_i L \frac{C_2 - C_3}{z_3^m - z_2^m} \tag{14}$$

$$Q_{\text{vert}}^{3,\text{bkgd}} = \sigma_w \kappa H \frac{l_c}{l_c + \kappa H} W_i L \frac{C_3 - C_{\text{bkgd}}}{z_{\text{bkgd}}^m - z_3^m} \tag{15}$$

with $C_1$, $C_2$, $C_3$ and $C_{\text{bkgd}}$ the concentrations of the three levels and the background respectively.

### 2.2.2 Mean horizontal wind speed in the street

As for the vertical turbulent flux, only the *Wang* parameterization among the three parameterizations available in MUNICH can be used to compute the mean horizontal wind speeds in the street. It is thanks to its explicit vertical dependency and the no-slip condition at the ground that is always satisfied (Maison et al., 2022). Therefore, the mean horizontal wind speed can be computed at each level in the street, by modifying Eq. 6 to integrate vertically between the level heights:





$$u_1 = \frac{1}{z_1 - z_{0s}} \int\limits_{z_{0s}}^{z_1} u_Y(z)\,\mathrm{d}z \qquad (16)$$

$$u_2 = \frac{1}{z_2 - z_1} \int\limits_{z_1}^{z_2} u_Y(z)\,\mathrm{d}z \qquad (17)$$

$$u_3 = \frac{1}{H - z_2} \int\limits_{z_2}^{H} u_Y(z)\,\mathrm{d}z. \qquad (18)$$

with $z_1$ and $z_2$ the limits of the first two levels as presented in Fig. 2.

## 3 Application to street networks in Copenhagen with comparison to OSPM

This section presents two applications of MUNICH-hete to assess its capabilities compared to MUNICH-homo and OSPM. Simulations are performed over the year 2019 for two street networks in Copenhagen, Denmark. The first street network is centered around the H. C. Andersens Boulevard and the second around the Jagtvej street. They are named HCAB and JGTV respectively in the following. They have been selected as observational data of CO, $NO_2$, $NO_x$ and $O_3$ are available for the HCAB and, $NO_2$ and $NO_x$ for JGTV over the whole year. OSPM simulations were also performed to compare model performances. As OSPM does not represent air fluxes at intersections, OSPM simulations are performed only for the streets where there are observations. Thanks to its coupled approach between a Gaussian plume model and a box model, OSPM is able to calculate concentrations on the two sides of the street (Berkowicz et al., 1997; Berkowicz, 2000a; Ottosen et al., 2015). Two receptors are used to compare with the observed concentrations on each side of the street: OSPM-R1 and OSPM-R2. The height of the receptors is taken as 2 m to correspond roughly to the height at which observations were performed. They are compared to the concentrations simulated at the first two levels of MUNICH-hete, which are representative of the concentrations at 1 m and 3 m.

MUNICH and OSPM use the same input data to estimate the street concentrations. The meteorological parameters originate from WRF simulations (Skamarock et al., 2008) and the background concentrations are simulated using the Urban Background Model (UBM) (Berkowicz, 2000b). Traffic emissions are generated using the OML-Highway model (Olesen et al., 2015) allowing for precise information for each street segment.

OSPM represents $NO_2$ and $O_3$ chemical transformations using a system of 2 reactions (Berkowicz et al., 1997; Berkowicz, 2000a). The first one describes the production of $NO_2$ due to reaction of NO with $O_3$, and the second one the photodissociation of $NO_2$ leading to reproduction of NO and $O_3$. For a fair comparison, MUNICH is configured to run with a simple chemistry





scheme, the Leighton photostationary state for $O_3$ (Leighton, 1961; Kim et al., 2018):

$$200 \qquad NO_2 + h\nu \quad \rightarrow \quad NO + O(^3P) \tag{R1}$$

$$O(^3P) + O_2 + M \quad \rightarrow \quad O_3 + M \tag{R2}$$

$$NO + O_3 \quad \rightarrow \quad NO_2 + O_2 \tag{R3}$$

### 3.1   H. C. Andersens Boulevard

H. C. Andersens Boulevard is a wide, densely-trafficked boulevard, with an aspect ratio $a_r = H/W$ of about 0.2. It is open
on one side with trees instead of buildings. This configuration can be represented in OSPM. However, in MUNICH, a mean
building height is defined for each street. Here, it is estimated by averaging the building and the tree heights. The simulated
street network is composed of 86 street segments centered around the street where the observation station is located (see
brown cross on Fig. 4). Figure 3 presents the monthly-averaged concentrations of CO and $NO_2$ from the OSPM and MUNICH
simulations compared to observations. Appendix C contains monthly-averaged concentrations of $NO_x$ and $O_3$, and statistical
indicators of the comparison for all four pollutants.

OSPM-R1 is the receptor that is close to the measurement station, thus better suited to be compared to observations. Dif-
ferences in concentrations between OSPM-R1 and OSPM-R2 highlight the importance to take into account horizontal hetero-
geneities in the street. Observations lies between the concentrations simulated at the two receptors, except for CO for which
OSPM slightly underestimates concentrations in the first half of the year. This underestimation could be linked to underesti-
mation of sources other than traffic, e.g. biomass burning, at the regional scale. Overall, the concentrations are well estimated
with OSPM with errors between 26% and 33% for CO, and between 34% and 46% for $NO_2$.

For CO, $NO_2$ and NOx, which are emitted by traffic in the bottom of the street, the concentrations are higher in the first level
MUNICH-hete-l1 near the bottom and lower in the third level near the roof level. For $O_3$, the opposite behavior is observed
as it is mainly imported by the atmosphere above the street, and it is titrated by NO near the ground. The first two levels
munich-hete-l1 and munich-hete-l2 have higher concentrations of CO, $NO_2$ and $NO_x$ than munich-homo, thus improving the
comparison to observations and OSPM concentrations. For CO, the error is improved from 36% in MUNICH-homo to 28%
in MUNICH-hete-l1, and for $NO_2$, it is improved from 48% in MUNICH-homo to 35% in MUNICH-hete-l1. Although the
concentrations of $NO_2$ and CO are underestimated in MUNICH-homo compared to observations, they are well modelled in
the first level MUNICH-hete-l1 with low errors and biases (see Appendix C). For CO, the concentrations simulated in the first
level MUNICH-hete-l1 lies between the two OSPM receptors for most of the year. This is the case for $NO_2$ as well, except
between April and August, when $NO_2$ concentrations are underestimated in MUNICH-hete. The formation of $NO_2$ in the lower
levels (see Reaction R3) could be limited with the current model setup, because volatile organic compounds are not taken into
account in the Danish cases. However, they may participate in $O_3$ and $NO_2$ formation through $HO_2$ and $RO_2$ radicals (Atkinson,
2000; Kwak and Baik, 2012; Zhong et al., 2017; Dai et al., 2021). Furthermore, the vertical discretization is coarse, limiting
$O_3$ transport deep into the street.



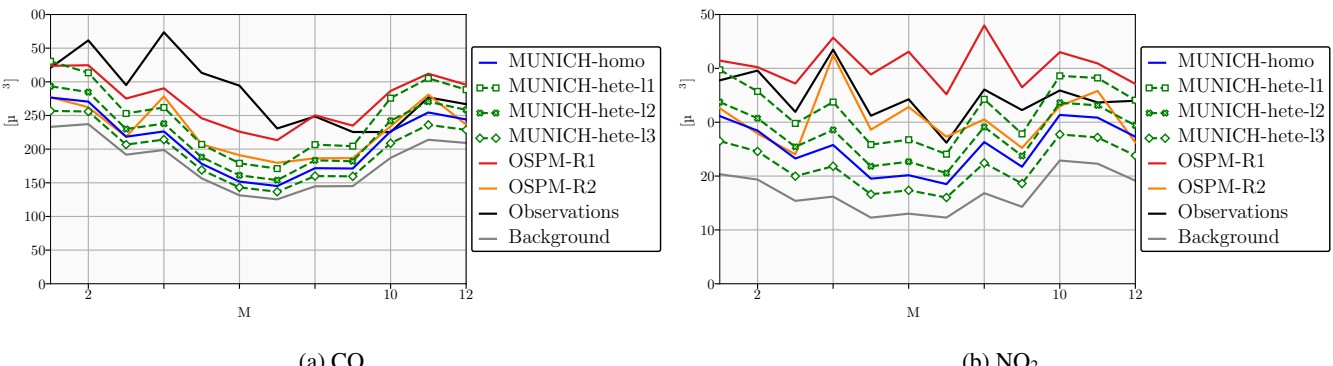

(a) CO

(b) NO$_2$

**Figure 3.** Monthly-average concentrations (in µg m$^{-3}$) of CO (a) and NO$_2$ (b) at HCAB monitoring station. The solid blue line represents the homogeneous version of MUNICH. The three green dashed lines represents the three levels of the heterogeneous version of MUNICH, the lowest level (l1) with square markers, the intermediate level (l2) with cross markers and the top level (l3) with diamond markers. The solid red line represents the OSPM receptor that is close to the measurement station and the solid orange line the second OSPM receptor located on the other side of the street. The observations are in solid black and the background concentrations in solid grey.

Over the whole street network, as presented in Fig. 4, the differences between the concentrations simulated in the first vertical level of MUNICH-hete compared those simulated in MUNICH-homo vary. For wide streets and avenue with dense traffic, the concentrations are higher in MUNICH-hete-l1 than in MUNICH-homo, with an increase by up to 23 % for CO and 30 % for NO$_2$. This increase is lower in more narrow and less frequented streets. Although the concentrations in MUNICH-hete-l1 are

235 always higher than those in MUNICH-homo for CO, for NO$_2$, in narrow streets, the concentrations are lower in MUNICH-hete-l1. These lower NO$_2$ concentrations are probably related to the limited transport of O$_3$ from the background to the bottom of the street, limiting the titration of NO.



(a) MUNICH-homo, CO concentration

(b) Relative differences of CO between
MUNICH-hete-l1 and MUNICH-homo

(c) MUNICH-homo, NO$_2$ concentration

(d) Relative differences of NO$_2$ between
MUNICH-hete-l1 and MUNICH-homo

**Figure 4.** CO and NO$_2$ time-averaged concentrations (in µg m$^{-3}$) for MUNICH-homo on the upper and lower left panels respectively for the HCAB street network. Relative differences (in %) between the first level of MUNICH-hete and MUNICH-homo for CO and NO$_2$ on the upper and lower right panels respectively for the HCAB street network. A positive relative difference indicates higher concentrations for MUNICH-hete. The brown cross on panels (a) and (c) represents the position of the measurement station.





## 3.2 Jagtvej

Jagtvej is a conventional street canyon with an aspect ratio of about 0.8. The simulated street network is composed of 265 street
segments centered around the street where the observation station is located (see brown cross on Fig. 6). The monthly-averaged
concentrations of CO and $NO_2$ from the OSPM and MUNICH simulations compared to observations are presented in Fig. 5.
Appendix D contains monthly-averaged concentrations of $NO_x$ and $O_3$, and statistical indicators of the comparison for $NO_2$
and $NO_x$.

OSPM-R2 is the receptor that is close to the measurement station, thus better suited to be compared to observations. The
differences between the concentration of the two OSPM receptors are lower in JGTV than in HCAB (see Fig. 3). JGTV is
narrower with higher buildings on both sides, thus limiting the ventilation zone and the penetration in the street of background
concentrations that would reduce the concentrations at the downwind receptor. OSPM tends to slightly overestimate $NO_2$ and
$NO_x$ concentrations (with errors between 41% and 53% and between 42% and 63% respectively).

Concentrations from the homogeneous version of MUNICH are close to OSPM concentrations for CO. They are lower
than OSPM concentrations for $NO_2$, but they compare well to observations with a bias of -3%, and an error of 46%. These
lower $NO_2$ concentrations are linked to lower $NO_x$ concentrations (with a bias of -19% compared to observations and an error
of 53%). As for HCAB, the MUNICH-hete concentrations decrease from the bottom to the top of the street. For CO and
$NO_x$, only the first level have concentrations higher than MUNICH-homo while for $NO_2$, all three levels have concentrations
lower than MUNICH-homo. $O_3$ concentrations are also higher for the three levels (see Fig. D1). The $NO_x$ concentrations of
MUNICH-hete-l1 compare slightly better to observations than MUNICH-homo, while MUNICH-homo is slightly better for
$NO_2$. However, the statistics of the two models are quite close (Appendix D).

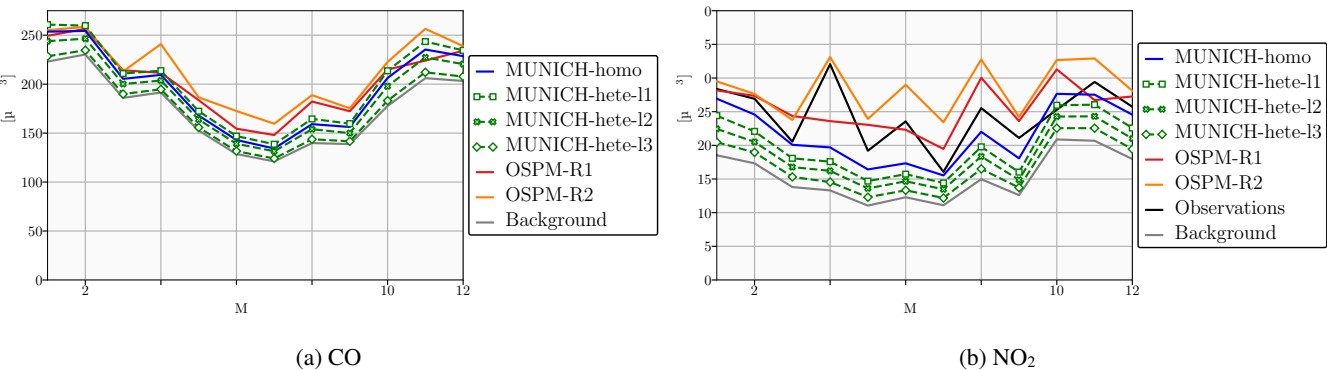

(a) CO                                                                                        (b) $NO_2$

**Figure 5.** Monthly-average concentrations (in µg m$^{-3}$) of CO (a) and $NO_2$ (b) at JGTV monitoring station. The solid blue line represents the
homogeneous version of MUNICH. The three green dashed lines represents the three levels of the heterogeneous version of MUNICH, the
lowest level (l1) with square markers, the intermediate level (l2) with cross markers and the top level (l3) with diamond markers. The solid
orange line represents the OSPM receptor that is close to the measurement station and the solid red line the second OSPM receptor located
on the other side of the street. The observations (only available for $NO_2$) are in solid black and the background concentrations in solid grey





Conclusions are similar for the whole street network (see Fig. 6). The CO concentrations are slightly higher in the first level of MUNICH-hete than in MUNICH-homo. However, $NO_2$ concentrations at the bottom of the street in the first level of MUNICH-hete tend to be lower than in MUNICH-homo. In some specific street segments of the network, the differences of concentrations for both CO and $NO_2$ are higher in MUNICH-hete-l1 than in MUNICH-homo. This is due to a mix of different traffic emissions and street morphologies which favor the transport of pollutants.

(a) MUNICH-homo, CO concentration

(b) Relative differences of CO between MUNICH-hete-l1 and MUNICH-homo

(c) MUNICH-homo, $NO_2$ concentration

(d) Relative differences of $NO_2$ between MUNICH-hete-l1 and MUNICH-homo

**Figure 6.** CO and $NO_2$ time-averaged concentrations (in $\mu g\,m^{-3}$) for MUNICH-homo in the upper and lower left panels respectively for the JGTV street network. Relative differences (in %) between the first level of MUNICH-hete and MUNICH-homo for CO and $NO_2$ in the upper and lower right panels respectively for JGTV. A positive relative difference indicates higher concentrations for MUNICH-hete. The brown cross on panels (a) and (c) represents the position of the measurement station.





## 4   Application to a street network in Greater Paris

The impacts of the discretization on gas and particle concentrations is evaluated over the street network near Paris, France, which was used to validate MUNICH v2.0 (Kim et al., 2022) and in several sensitivity studies (Lugon et al., 2021b; Sarica

et al., 2022). The street network represents a district of Le Perreux-sur-Marne, a suburb 13 km east of Paris, France. It is composed of 577 street segments (see Fig. 8). Simulations are performed from 22 March to 15 June 2014 with input data (emissions, meteorological parameters and background concentrations) from the reference simulation SCN0 of Sarica et al. (2022). Observational data are available for the whole simulation period for $NO_2$, $NO_x$, $PM_{2.5}$, $PM_{10}$ and black carbon (BC) at a segment of Boulevard d'Alsace Lorraine (see brown cross on Fig. 8). They were performed at a height of about 2 m and they

are thus to be compared to the concentrations of the first two levels (i=1 and i=2) of MUNICH-hete.

The average daily profiles of $NO_x$ and $PM_{10}$ concentrations are shown in Fig. 7, and for NO, $NO_2$ and the statistical indicators in Appendix E1. As in the Danish streets, for pollutants emitted by traffic, the concentrations are higher in MUNICH-hete at the bottom of the street decreasing to the roof level. The concentrations of $PM_{10}$, $NO_2$, $NO_x$ and BC are higher on average by 12 %, 21 %, 40 % and 30 % respectively in MUNICH-hete-l1 than in MUNICH-homo. The lower concentration difference for $PM_{10}$

than for the other compounds reflects that non-traffic sources are more important for $PM_{10}$, and inversely they are small for BC. Despite the higher concentrations in MUNICH-hete-l1, BC concentrations remain strongly underestimated compared to observations, in agreement with the CFD simulations of Lin et al. (2022). For $PM_{10}$, $NO_2$ and $NO_x$, the concentrations compare well to observations, e.g. the error is 33% for $NO_2$ and 37% for $PM_{10}$, with slightly better statistics using MUNICH-hete-l1 than MUNICH-homo. For all pollutants, the concentrations of the intermediate level (i=2) are very close to the concentrations

of MUNICH-homo. This is due to a mixture of different parameters such as street morphology and dispersion conditions over the simulated period.

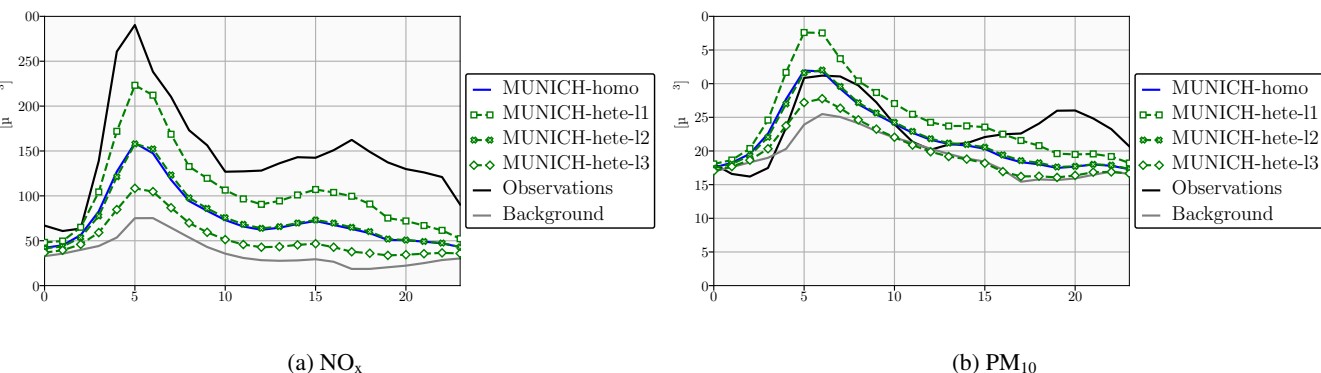

(a) $NO_x$          (b) $PM_{10}$

**Figure 7.** Average daily profile of the concentrations of $NO_x$ (a) and $PM_{10}$ (b) over the simulation period at the Boulevard d'Alsace Lorraine monitoring station. The solid blue line represents the homogeneous version of MUNICH. The three green dashed lines represents the three levels of the heterogeneous version of MUNICH, the lowest level (l1) with square markers, the intermediate level (l2) with cross markers and the top level (l3) with diamond markers. The observations are in solid black and the background concentrations in solid grey





In streets, both BC and organic matter (OM) exhibit higher concentrations than in the background (Lugon et al., 2021a). OM consists of primary and secondary aerosols that are formed from the oxidation of volatile organic compounds and/or the condensation of semi-volatile organic compounds. Concentrations of both BC and OM simulated in MUNICH-homo and in the first level of MUNICH-hete are compared in Fig 8 (see Appendix Fig. E2 for $NO_2$ and $PM_{10}$). The concentrations simulated in MUNICH-hete-l1 are always higher than in MUNICH-homo. In most streets of the network that are narrow and with limited traffic, the increase in concentrations is limited. It is more important for more open and frequented streets, such as the Boulevard d'Alsace Lorraine.

BC being a primary inert pollutant emitted by traffic, its concentrations are strongly influenced by the discretization as emissions are no longer artificially diluted in the whole street volume. They are now constraint at the bottom of the street inducing an average increase of 30 % compared to the homogeneous version. For OM and $PM_{10}$, the increase is lower (16 % and 12 % respectively), because of the stronger influence of non-traffic sources.





(a) MUNICH-homo, OM concentration

(b) Relative differences of OM between
MUNICH-hete-l1 and MUNICH-homo

(c) MUNICH-homo, BC concentration

(d) Relative differences of BC between
MUNICH-hete-l1 and MUNICH-homo

**Figure 8.** OM and BC time-averaged concentrations (in µg m$^{-3}$) for MUNICH-homo in the upper and lower left panels respectively for the district of Le Perreux-sur-Marne. Relative differences (in %) between the first level of MUNICH-hete and MUNICH-homo for OM and BC in the upper and lower right panels respectively for the district of Le Perreux-sur-Marne. A positive relative difference indicates higher concentrations for MUNICH-hete. The brown cross on panels (a) and (c) represents the position of the measurement station.





## 5 Sensitivity analysis

In this section, the sensitivity of the heterogeneous version of MUNICH to the presence of a street network, which influences
the concentrations entering the street via the upwind intersection, is estimated.

The computation of inflow and outflow fluxes at intersections is performed by estimating the balance of fluxes entering and
leaving the intersection from the different street segments attached to it. If this balance is not perfect, there are exchanges with
the atmosphere above the intersection. When the total flux entering the intersection is higher than the one leaving it, the flow
overload is directed to the atmosphere. When the total flux leaving the intersection is higher, a flux from the atmosphere to the
300 intersection is considered. Further explanation is available in Kim et al. (2018, 2022).

Without a street network around the street segment of interest, the pollutant mass fluxes entering the street are determined
from the background concentration. The contribution of the neighboring streets is thus not taken into account and concentra-
tions of pollutants emitted in streets are expected to be lower than when there is a street network.

Table 1 and App. F present the influence of the neighboring streets for the three cases of the study. It is quantified using the
305 normalized mean error (NME) and bias (NMB) between the simulations without and with the neighboring streets. As expected,
without the street network, the concentrations are lower. The bias is between 12% and 21% for $NO_2$, 18% and 27% for $NO_x$
and 14% for BC. Biases are lower for OM and PM (2 %), because of stronger influence of background concentrations for those
compounds.

**Table 1.** Statistical indicators of the influence of the street network on concentrations simulated with MUNICH-hete for the street segment
of the Boulevard d'Alsace Lorraine with the monitoring station. Indicators are presented in Appendix B.

| MUNICH-hete level | | | $NO_2$ | $NO_x$ | $PM_{10}$ | $PM_{2.5}$ | BC | OM |
|---|---|---|---|---|---|---|---|---|
| l1 | With network | Mean concentration [$\mu g\,m^{-3}$] | 58.92 | 104.74 | 24.58 | 22.18 | 1.84 | 6.13 |
| | Without network | Mean concentration [$\mu g\,m^{-3}$] | 51.50 | 85.46 | 24.15 | 21.97 | 1.58 | 5.98 |
| | | NME [%] | 23.25 | 23.63 | 17.26 | 17.28 | 20.95 | 20.29 |
| | | NMB [%] | -12.58 | -18.41 | -1.75 | -0.98 | -14.02 | -2.37 |
| l2 | With network | Mean concentration [$\mu g\,m^{-3}$] | 47.81 | 75.36 | 21.94 | 20.08 | 1.42 | 5.30 |
| | Without network | Mean concentration [$\mu g\,m^{-3}$] | 42.67 | 62.34 | 21.77 | 20.04 | 1.25 | 5.25 |
| | | NME [%] | 20.97 | 23.11 | 14.55 | 14.46 | 19.37 | 17.11 |
| | | NMB [%] | -10.74 | -17.28 | -0.75 | -0.20 | -12.15 | -0.96 |
| l3 | With network | Mean concentration [$\mu g\,m^{-3}$] | 37.97 | 52.63 | 19.91 | 18.41 | 1.09 | 4.69 |
| | Without network | Mean concentration [$\mu g\,m^{-3}$] | 35.65 | 47.01 | 19.89 | 18.44 | 1.02 | 4.69 |
| | | NME [%] | 12.83 | 15.21 | 8.11 | 8.05 | 11.78 | 9.47 |
| | | NMB [%] | -6.10 | -10.67 | -0.08 | 0.15 | -6.70 | -0.12 |



## 6 Conclusions

The street-network model MUNICH v2.0 has been modified to introduce concentration heterogeneities in the street, and to better represent population exposure. To model the vertical gradients frequently observed, the streets were discretized with three levels, thus limiting the artificial dilution of emissions and concentrations. Based on a parameterization from OSPM, a ventilation zone is considered under specific conditions to represent horizontal heterogeneities. In order to test these developments, the heterogeneous version of MUNICH (MUNICH-hete) has been applied to two cases in Copenhagen, Denmark, with

comparisons to OSPM, and to one case near Paris, France. Overall, MUNICH-hete improves the comparison to observations compared to the homogeneous version. The errors to observations are reduced by up to 20 % for $NO_x$ and 15 % for BC.

As expected, in MUNICH-hete, concentrations of compounds emitted by traffic (CO, $NO_2$, $NO_x$, $PM_{10}$, BC and OM) are higher at the bottom of the street than at the top. These increases can reach up to 60 % and 30 % for $NO_2$ and $PM_{10}$ respectively. The intermediate level, serving as a buffer, presents concentrations higher or similar to the homogeneous version (MUNICH-

320 homo). Finally, concentrations in the highest level, in direct contact with the atmosphere above the street, are the lowest of the street. For the Danish cases, the low $NO_2$ concentrations observed in the lower levels could be related to the absence of volatile organic compounds in the model setup and the coarse vertical discretization limiting $O_3$ transport deep into the street.

A sensitivity study of the influence of the street network on concentrations in the streets shows the importance of considering neighboring streets in MUNICH. When no network is considered, concentrations in the street are lower due to the overestimated

impact of the atmosphere above. At the bottom of the street, concentrations of $NO_2$ and BC are reduced by up to 28 % and 14 % respectively without a network. PM and OM are less impacted with a reduction of about 2 % due to a strong influence of non-traffic sources.

For the next step, the ventilation zone will be fully discretized vertically to facilitate the penetration of background concentrations to the bottom of the street. The horizontal exchange fluxes between the two zones will be also be modelled. Deposition

and resuspension processes that were not considered in this development will be added. Finally, the fluxes that are currently assumed to be vertically homogeneous will be discretized.

*Code and data availability.* MUNICH-hete is available at Sarica et al. (2023). The configuration files, the input data and also the scripts to generate the figures and statistics are available at Sarica et al. (2023).





## Appendix A: Volumes of the recirculation and ventilation zones

The algorithm used in the heterogeneous version of MUNICH to compute the volumes of the recirculation and ventilation zones
is based on the parameterization of OSPM (Berkowicz et al., 1997; Berkowicz, 2000a; Ottosen et al., 2015). This algorithm
is applied at the beginning of each time step as the size of the recirculation zone is dependent on wind speed and direction.
Fig A1 presents the shape of the recirculation zone and the associated parameters.

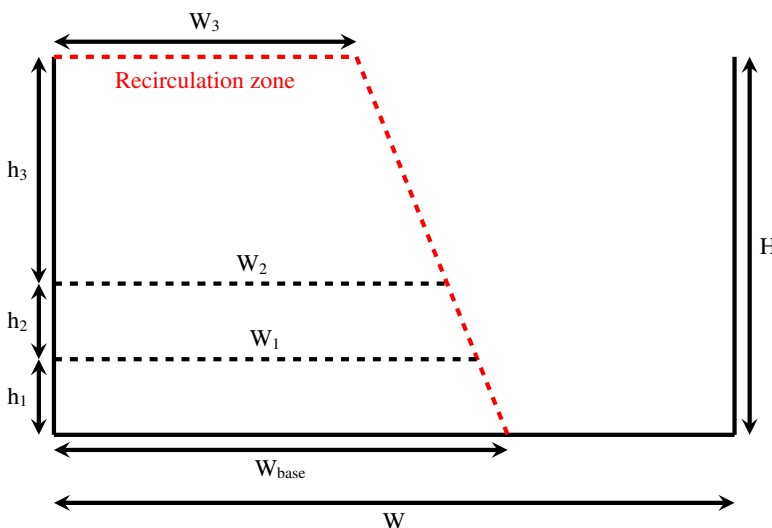

**Figure A1.** Representation of the recirculation zone in the heterogeneous version of MUNICH.

The first step is to compute the length of the vortex in the direction of the wind:

$$L_{\mathrm{vortex}} = 2Hf \tag{A1}$$

with $H$ the height of the street and

$$f = \begin{cases} 1 \; if \; u_{\mathrm{roof}} \; \geq \; 2ms^{-1} \\ \sqrt{0.5u_{\mathrm{roof}}} \; if \; u_{\mathrm{roof}} \; < \; 2ms^{-1} \end{cases} \tag{A2}$$

with $u_{\mathrm{roof}}$ the wind speed at roof level.

The width of the trapeze base is the projection of $L_{\mathrm{vortex}}$ in the street:

$$W_{base} = L_{\mathrm{vortex}} sin(\theta)) \tag{A3}$$

with $\theta$ the angle between the wind direction and the street orientation.

The width of the trapeze top is equal to half of the base:

$$W_3 = \frac{L_{\mathrm{vortex}} sin(\theta)}{2} \tag{A4}$$





Knowing these lengths and using algebraic considerations, the widths $W_1$ and $W_2$ can be calculated:

$$
\begin{cases}
W_1 = W_3 + \dfrac{(h_2 + h_3)\Delta W}{H} \\[2mm]
W_2 = W_3 + \dfrac{h_3 \Delta W}{H}
\end{cases}
\tag{A5}
$$

with $\Delta W = W_{base} - W_3$

The horizontal surfaces for vertical exchanges between the levels and with the concentrations above the street are calculated as followed:

$$
\begin{cases}
S_1^h = W_1 L \\
S_2^h = W_2 L \\
S_3^h = W_3 L
\end{cases}
\tag{A6}
$$

with $L$ the street length.

The vertical surfaces for advection via intersections are determined with:

$$
\begin{cases}
S_1^v = \dfrac{W_1 (h_1)^2 (W_{base} - W_1)}{2} \\[2mm]
S_2^v = \dfrac{W_2 (h_2)^2 (W_1 - W_2)}{2} \\[2mm]
S_3^v = \dfrac{W_3 (h_3)^2 (W_2 - W_3)}{2}
\end{cases}
\tag{A7}
$$

Finally, the volumes associated to each level of the recirculation zone are :

$$
\begin{cases}
V_1 = h_1 L \left( W_1 + \dfrac{(W_{base} - W_1)}{2} \right) \\[2mm]
V_2 = h_2 L \left( W_2 + \dfrac{(W_1 - W_2)}{2} \right) \\[2mm]
V_3 = h_3 L \left( W_3 + \dfrac{(W_2 - W_3)}{2} \right)
\end{cases}
\tag{A8}
$$

In the current version of MUNICH-hete, it is assumed that traffic emissions are all affected to the recirculation zone. Therefore, its base $W_{base}$ has to be superior or equal to the street width $W$. If it is not the case, the recirculation zone is considered to fill the whole street volume.

When considered, the other widths are also limited by the street width:

$$
\begin{cases}
W_1 = min(W, W_1) \\
W_2 = min(W, W_2) \\
W_3 = min(W, W_3)
\end{cases}
\tag{A9}
$$





## Appendix B: Statistical indicators

For evaluation of the simulations to observations, the following statistical indicators are used. $o$ and $s$ represent the observed and the simulated concentrations respectively. The overbar represents the average.

- Mean fractional error (MFE):

$$MFE = 2 \times \overline{\left( \frac{|s-o|}{s+o} \right)}$$

- Mean fractional bias (MFB):

$$MFB = 2 \times \overline{\left( \frac{s-o}{s+o} \right)}$$

- Factor of 2 (FAC2): fraction of data that satisfy $0.5 \leq \frac{s}{o} \leq 2.0$

For comparison of the simulations, the normalized mean bias (NMB) and normalized mean error (NME) are used. X represents concentrations with $X_0$ the reference simulation and $X_i$ the compared simulation. The overbar represents the average.

- Normalized mean bias:

$$NMB = \frac{\overline{(X_i - X_0)}}{\overline{X_0}}$$

- Normalized mean error:

$$NME = \frac{\overline{|X_i - X_0|}}{\overline{X_0}}$$





## Appendix C: Additional information for HCAB

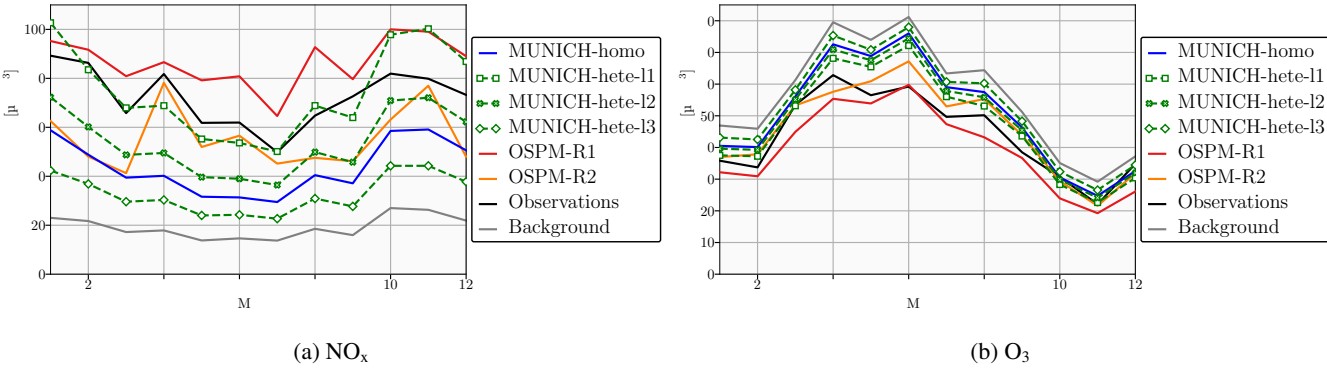

(a) NO$_x$  (b) O$_3$

**Figure C1.** Monthly-average concentrations (in µg m$^{-3}$) of NO$_x$ (a) and O$_3$ (b) at HCAB monitoring station. The solid blue line represents the homogeneous version of MUNICH. The three green dashed lines represents the three levels of the heterogeneous version of MUNICH, the lowest level (l1) with square markers, the intermediate level (l2) with cross markers and the top level (l3) with diamond markers. The solid red line represents the OSPM receptor that is close to the measurement station and the solid orange line the second OSPM receptor located on the other side of the street. The observations are in solid black and the background concentrations in solid grey



**Table C1.** Statistical indicators of the evaluation of the hourly simulated concentrations to observations at HCAB monitoring station. Indicators are presented in Appendix B.

| | | CO | $NO_2$ | $NO_x$ | $O_3$ |
|---|---|---|---|---|---|
| Observations | Mean concentration [µg m$^{-3}$] | 285.35 | 34.64 | 72.30 | 44.33 |
| MUNICH-homo | Mean concentration [µg m$^{-3}$] | 210.88 | 25.34 | 43.77 | 50.44 |
| | MFE | 0.36 | 0.48 | 0.55 | 0.38 |
| | MFB | -0.29 | -0.29 | -0.42 | 0.14 |
| | FAC2 | 0.91 | 0.73 | 0.63 | 0.83 |
| MUNICH-hete-l1 | Mean concentration [µg m$^{-3}$] | 249.29 | 32.37 | 74.95 | 47.34 |
| | MFE | 0.28 | 0.35 | 0.35 | 0.37 |
| | MFB | -0.14 | -0.06 | 0.05 | 0.07 |
| | FAC2 | 0.96 | 0.87 | 0.86 | 0.84 |
| MUNICH-hete-l2 | Mean concentration [µg m$^{-3}$] | 223.39 | 27.69 | 53.86 | 49.36 |
| | MFE | 0.32 | 0.42 | 0.43 | 0.37 |
| | MFB | -0.24 | -0.21 | -0.24 | 0.12 |
| | FAC2 | 0.93 | 0.80 | 0.78 | 0.83 |
| MUNICH-hete-l3 | Mean concentration [µg m$^{-3}$] | 197.59 | 21.87 | 32.90 | 52.59 |
| | MFE | 0.39 | 0.57 | 0.73 | 0.39 |
| | MFB | -0.34 | -0.43 | -0.65 | 0.20 |
| | FAC2 | 0.88 | 0.61 | 0.42 | 0.82 |
| OSPM-R1 | Mean concentration [µg m$^{-3}$] | 272.86 | 40.59 | 86.60 | 39.50 |
| | MFE | 0.26 | 0.34 | 0.40 | 0.35 |
| | MFB | -0.05 | 0.19 | 0.21 | -0.11 |
| | FAC2 | 0.96 | 0.87 | 0.82 | 0.86 |
| OSPM-R2 | Mean concentration [µg m$^{-3}$] | 227.84 | 30.53 | 55.43 | 45.77 |
| | MFE | 0.33 | 0.46 | 0.61 | 0.36 |
| | MFB | -0.23 | -0.16 | -0.33 | 0.05 |
| | FAC2 | 0.93 | 0.74 | 0.61 | 0.84 |





## Appendix D: Additional information for JGTV

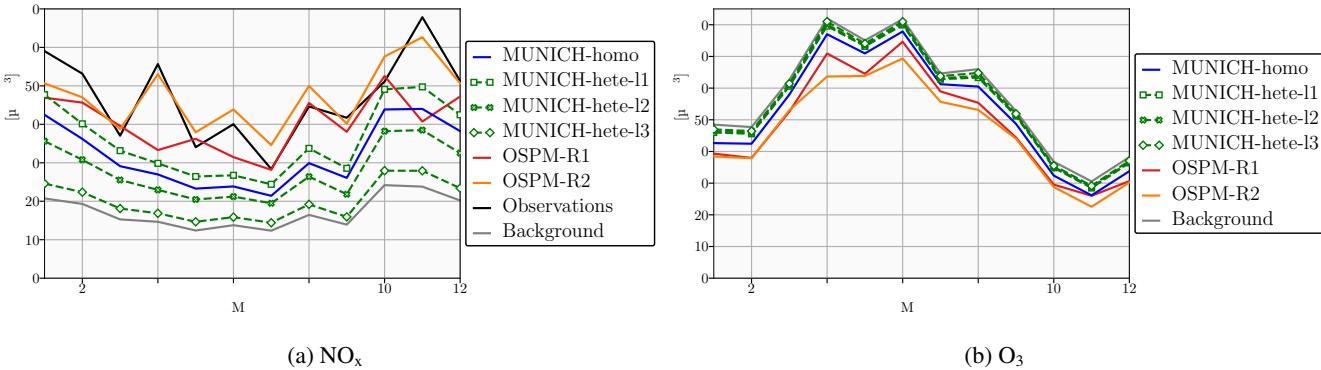

(a) NO$_x$                           (b) O$_3$

**Figure D1.** Monthly-average concentrations (in µg m$^{-3}$) of NO$_x$ (a) and O$_3$ (b) at JGTV monitoring station. The solid blue line represents the homogeneous version of MUNICH. The three green dashed lines represents the three levels of the heterogeneous version of MUNICH, the lowest level (l1) with square markers, the intermediate level (l2) with cross markers and the top level (l3) with diamond markers. The solid orange line represents the OSPM receptor that is close to the measurement station and the solid red line the second OSPM receptor located on the other side of the street. The observations are in solid black and the background concentrations in solid grey





**Table D1.** Statistical indicators of the evaluation of the hourly simulated concentrations to observations at JGTV monitoring station. Indicators are presented in Appendix B.

|  |  | NO$_2$ | NO$_x$ |
| --- | --- | --- | --- |
| Observations | Mean concentration [µg m$^{-3}$] | 24.50 | 46.94 |
| MUNICH-homo | Mean concentration [µg m$^{-3}$] | 21.69 | 32.10 |
|  | MFE | 0.46 | 0.53 |
|  | MFB | -0.03 | -0.19 |
|  | FAC2 | 0.75 | 0.66 |
| MUNICH-hete-l1 | Mean concentration [µg m$^{-3}$] | 19.77 | 35.99 |
|  | MFE | 0.49 | 0.51 |
|  | MFB | -0.13 | -0.11 |
|  | FAC2 | 0.75 | 0.68 |
| MUNICH-hete-l2 | Mean concentration [µg m$^{-3}$] | 18.37 | 27.78 |
|  | MFE | 0.52 | 0.58 |
|  | MFB | -0.20 | -0.32 |
|  | FAC2 | 0.71 | 0.61 |
| MUNICH-hete-l3 | Mean concentration [µg m$^{-3}$] | 16.81 | 20.10 |
|  | MFE | 0.57 | 0.74 |
|  | MFB | -0.29 | -0.58 |
|  | FAC2 | 0.68 | 0.46 |
| OSPM-R1 | Mean concentration [µg m$^{-3}$] | 25.60 | 40.46 |
|  | MFE | 0.53 | 0.63 |
|  | MFB | 0.15 | 0.03 |
|  | FAC2 | 0.67 | 0.56 |
| OSPM-R2 | Mean concentration [µg m$^{-3}$] | 28.41 | 47.21 |
|  | MFE | 0.41 | 0.42 |
|  | MFB | 0.25 | 0.18 |
|  | FAC2 | 0.80 | 0.79 |



## Appendix E:  Additional information for the district of Le Perreux-sur-Marne

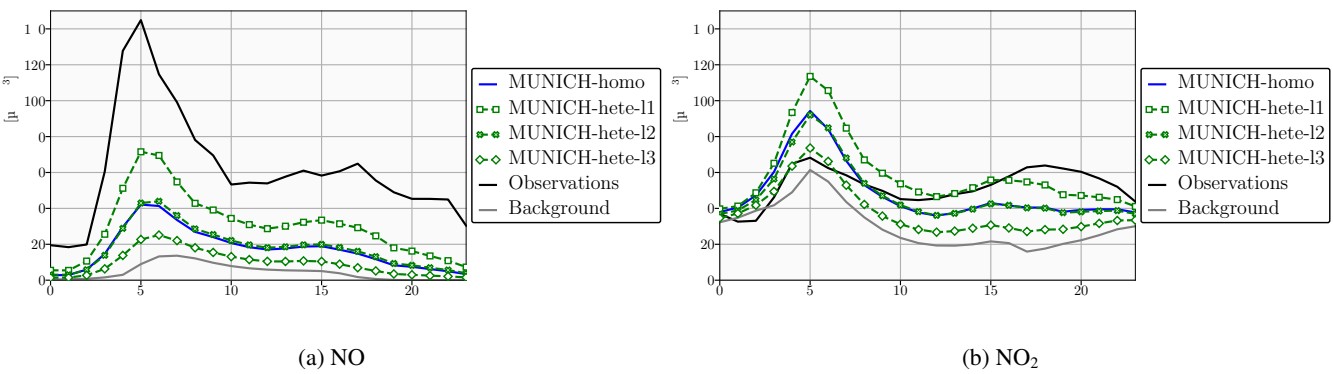

(a) NO                                                    (b) NO$_2$

**Figure E1.** Average daily profile of the concentrations of NO (a) and NO$_2$ (b) over the simulation period at the Boulevard d'Alsace Lorraine monitoring station. The solid blue line represents the homogeneous version of MUNICH. The three green dashed lines represents the three levels of the heterogeneous version of MUNICH, the lowest level (l1) with square markers, the intermediate level (l2) with cross markers and the top level (l3) with diamond markers. The observations are in solid black and the background concentrations in solid grey





**Table E1.** Statistical indicators of the evaluation of the hourly simulated concentrations to observations at the Boulevard d'Alsace Lorraine monitoring station. Indicators are presented in Appendix B.

|  |  | $NO_2$ | $NO_x$ | $PM_{10}$ | $PM_{2.5}$ | BC |
|---|---|---|---|---|---|---|
| Observations | Mean concentration [μg m$^{-3}$] | 52.16 | 147.19 | 23.42 | 12.72 | 6.10 |
| MUNICH-homo | Mean concentration [μg m$^{-3}$] | 48.49 | 74.62 | 21.89 | 20.06 | 1.41 |
|  | MFE | 0.36 | 0.65 | 0.38 | 0.53 | 1.17 |
|  | MFB | -0.11 | -0.60 | -0.08 | 0.38 | -1.16 |
|  | FAC2 | 0.87 | 0.52 | 0.84 | 0.67 | 0.09 |
| MUNICH-hete-l1 | Mean concentration [μg m$^{-3}$] | 58.92 | 104.74 | 24.58 | 22.18 | 1.84 |
|  | MFE | 0.33 | 0.41 | 0.37 | 0.57 | 1.01 |
|  | MFB | 0.08 | -0.30 | 0.03 | 0.47 | -1.00 |
|  | FAC2 | 0.91 | 0.80 | 0.86 | 0.61 | 0.16 |
| MUNICH-hete-l2 | Mean concentration [μg m$^{-3}$] | 47.81 | 75.36 | 21.94 | 20.08 | 1.42 |
|  | MFE | 0.36 | 0.64 | 0.38 | 0.53 | 1.17 |
|  | MFB | -0.12 | -0.59 | -0.08 | 0.38 | -1.16 |
|  | FAC2 | 0.86 | 0.53 | 0.84 | 0.67 | 0.08 |
| MUNICH-hete-l3 | Mean concentration [μg m$^{-3}$] | 37.97 | 52.63 | 19.91 | 18.41 | 1.09 |
|  | MFE | 0.49 | 0.90 | 0.42 | 0.50 | 1.30 |
|  | MFB | -0.35 | -0.88 | -0.18 | 0.30 | -1.30 |
|  | FAC2 | 0.71 | 0.25 | 0.80 | 0.70 | 0.06 |



(a) MUNICH-homo, NO$_2$ concentration

(b) Relative differences of NO$_2$ between
MUNICH-hete-l1 and MUNICH-homo

(c) MUNICH-homo, PM$_{10}$ concentration

(d) Relative differences of PM$_{10}$ between
MUNICH-hete-l1 and MUNICH-homo

**Figure E2.** NO$_2$ and PM$_{10}$ time-averaged concentrations (in µg m$^{-3}$) for MUNICH-homo in the left panel for the district of Le Perreux-sur-Marne. Relative differences (in %) between the first level of MUNICH-hete and MUNICH-homo for PM$_{10}$ in the right panel for the district of Le Perreux-sur-Marne. A positive relative difference indicates higher concentrations for MUNICH-hete. The brown cross on panels (a) and (c) represents the position of the measurement station.





## Appendix F: Additional information for the sensitivity analysis

**Table F1.** Statistical indicators of the influence of the street network on concentrations simulated with MUNICH-hete for the street segment HCAB with the monitoring station. Indicators are presented in Appendix B.

| MUNICH-hete level | | | CO | NO$_2$ | NO$_x$ | O$_3$ |
|---|---|---|---|---|---|---|
| l1 | With network | Mean concentration [µg m$^{-3}$] | 249.29 | 32.37 | 74.95 | 47.34 |
| | Without network | Mean concentration [µg m$^{-3}$] | 214.27 | 23.28 | 46.69 | 52.98 |
| | | NME [%] | 14.05 | 28.06 | 37.70 | 11.90 |
| | | NMB [%] | -14.05 | -28.06 | -37.70 | 11.90 |
| l2 | With network | Mean concentration [µg m$^{-3}$] | 223.39 | 27.69 | 53.86 | 49.36 |
| | Without network | Mean concentration [µg m$^{-3}$] | 198.12 | 20.78 | 33.47 | 53.79 |
| | | NME [%] | 11.32 | 24.95 | 37.86 | 8.99 |
| | | NMB [%] | -11.32 | -24.95 | -37.86 | 8.99 |
| l3 | With network | Mean concentration [µg m$^{-3}$] | 197.59 | 21.87 | 32.90 | 52.59 |
| | Without network | Mean concentration [µg m$^{-3}$] | 186.96 | 18.65 | 24.33 | 54.78 |
| | | NME [%] | 5.38 | 14.73 | 26.05 | 4.17 |
| | | NMB [%] | -5.38 | -14.73 | -26.05 | 4.17 |

**Table F2.** Statistical indicators of the influence of the street network on concentrations simulated with MUNICH-hete for the street segment JGTV with the monitoring station. Indicators are presented in Appendix B.

| MUNICH-hete level | | | CO | NO$_2$ | NO$_x$ | O$_3$ |
|---|---|---|---|---|---|---|
| l1 | With network | Mean concentration [µg m$^{-3}$] | 201.21 | 19.77 | 35.99 | 55.19 |
| | Without network | Mean concentration [µg m$^{-3}$] | 189.24 | 17.46 | 27.56 | 56.45 |
| | | NME [%] | 6.09 | 12.38 | 23.63 | 3.13 |
| | | NMB [%] | -5.95 | -11.70 | -23.43 | 2.29 |
| l2 | With network | Mean concentration [µg m$^{-3}$] | 189.49 | 18.37 | 27.78 | 55.54 |
| | Without network | Mean concentration [µg m$^{-3}$] | 181.01 | 16.52 | 21.80 | 56.64 |
| | | NME [%] | 4.71 | 10.87 | 21.85 | 2.85 |
| | | NMB [%] | -4.47 | -10.03 | -21.53 | 1.99 |
| l3 | With network | Mean concentration [µg m$^{-3}$] | 178.59 | 16.81 | 20.10 | 56.12 |
| | Without network | Mean concentration [µg m$^{-3}$] | 176.03 | 15.92 | 18.30 | 56.80 |
| | | NME [%] | 1.84 | 6.16 | 9.57 | 1.83 |
| | | NMB [%] | -1.43 | -5.30 | -8.92 | 1.21 |



*Author contributions.* TS, AM, KS and YR conceptualized the development. TS, KS and YK developed the software and TS performed the simulations. MK and SSJ provided help for simulations with OSPM and the data for the Danish cases. TS, KS and YR performed the analysis. KS, YR, MK and SSJ supervised the study. KS and CC were responsible for funding acquisition. TS and KS wrote the original draft, all authors reviewed and edited the final manuscript.

*Competing interests.* The authors declare that they have no conflict of interest.





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
