# Peer review of "Modelling concentration heterogeneities in streets using the street-network model MUNICH"

_Geoscientific Model Development, 2023_

## Referee Comment (RC2)

**Referee Comment on gmd-2023-70 "Modelling concentration heterogeneities in streets using the street-network model MUNICH" by Thibaud Sarica et al.**

General Comment

A new version of the street-network model MUNICH is presented that introduces a vertical discretization of street canyons in three layers and the horizontal discretization of the street canyon in two zones (recirculation and ventilation) based on well-evaluated parameterizations of the OSPM model. The manuscript is well organized although more discussion of pertinent issues such as the influence of the vehicle wake and atmospheric stability on the vertical pollutant distribution in street canyons would give the paper more credibility. The new version, termed MUNICH-hete is applied to two street canyons and districts in Copenhagen and to one district in Greater Paris and compared to the homogeneous version of the model and OSPM results for two street-level receptors as well as street-level measurements. Further, a sensitivity study of the influence of the street network on the concentrations in the streets with MUNICH-hete was performed concluding that intersections with neighboring streets underline the need for simulating a street network rather than a single street.

MUNICH-hete predicts that concentrations of compounds emitted by traffic are higher at the bottom of the street than at the top. However, the vertical distribution of pollutants in the three-layer street canyon is not compared to measurements. For wide streets and avenues with dense traffic, the concentrations in the first level (MUNICH-hete-11) are generally higher than in the homogenous version of the model. A systematic analysis of the influence of the aspect ratio on the concentration differences between MUNICH-hete-11 and the homogenous version has unfortunately not been performed.

Specific Comments:

1.) P2, line 35-36: EPISODE-CityChem (Karl et al., 2019) that uses a simplified OSPM and CALIOPE-Urban (Benavides et al., 2019), which applies a downscaling depending on atmospheric stability and building density, should be mentioned here.

2.) P2, line 42: Was the SSH-aerosol model used in this study? It would be interesting to see the effect of detailed VOC chemistry on SOA concentrations along street canyons.

3.) P3, line 60: "The new version of MUNICH" should be given a version number. Is it based on MUNICH v2.0 in all other aspects than the vertical discretization or is it a different development branch?

4.) Resuspension of road dust can be an important of the non-exhaust particulate matter emissions from vehicles and its relevance will increase in the future due to transition to electric

mobility. Here it is stated that resuspension is a minor process compared to transport and chemistry, however that only applies to vehicular exhaust emissions. The consequence would be to restrict the application of MUNICH-hete to vehicular exhaust emissions of gaseous pollutants.

5.) How is the influence of atmospheric stability on vertical mixing within a street canyon taken into account in MUNICH? Some dispersion models like SIRANE account for this.

6.) Vertical distribution of pollutants in the street canyons of this study with MUNICH-hete should be shown, and compared to observation in different heights of the street canyons. There is some debate about the shape of the vertical profile in street canyons. Zoumakis et al. (1995) based on analysis of measurements report that the average vehicular pollutant concentration profile follows the general exponential form rather than the simple exponential function or Gaussian distribution. Kumar et al. (2009) report that concentrations of particle numbers increase from road level up to 2 m height, this increase is reproduced by a CFD model.

7.) This also concerns the question whether three levels are sufficient to represent the pollutant distribution in a street canyon. As I understand it, the height of the first level defines the volume in which traffic emissions are injected and diluted. The height of the first layer should be evaluated with a model that can resolve the flow and dispersion in the vehicle wake (Kumar et al., 2011). A sensitivity study on the height of the munich-11 level should be carried out.

8.) How is traffic-induced turbulence considered in the vertical turbulence flux sigma_w of Equation 12? The influence of the traffic-induced turbulence should then only be considered in the first level.

9.) P7, line 173: Explain no-slip condition at the ground.

10.) Is it planned to use 3-D building heights to inform the model on the real street canyon geometries?

Technical Corrections:

P 1, line 10: "Results show an improvement". In the abstract, give quantitative information on the improvement and state compared to which model the improvement was achieved.

P9, line 226: "when NO2 concentrations are underestimated" – does this refer to measured NO2?

Figure 3: Annotations on x-axis and y-axis are incomplete.

Figure 5: Same as for Figure 3. In Figure 5a the line for the CO observations is missing.

Figure 7: Same as for Figure 3.

References:

Benavides, J., Snyder, M., Guevara, M., Soret, A., Pérez García-Pando, C., Amato, F., Querol, X., and Jorba, O.: CALIOPE-Urban v1.0: coupling R-LINE with a mesoscale air quality modelling system for urban air quality forecasts over Barcelona city (Spain), Geosci. Model Dev., 12, 2811–2835, https://doi.org/10.5194/gmd-12-2811-2019, 2019.

Karl, M., Walker, S.-E., Solberg, S., and Ramacher, M. O. P.: The Eulerian urban dispersion model EPISODE – Part 2: Extensions to the source dispersion and photochemistry for EPISODE–CityChem v1.2 and its application to the city of Hamburg, Geosci. Model Dev., 12, 3357–3399, https://doi.org/10.5194/gmd-12-3357-2019, 2019.

Kumar, P., Garmory, A., Ketzel, M., Berkowicz, R., and Britter, R.: Comparative study of measured and modelled number concentrations of nanoparticles in an urban street canyon, Atmos. Environ., 43, 949–958, doi:10.1016/j.atmosenv.2008.10.025, 2009.

Kumar, P., Ketzel, M., Vardoulakis, S., Pirjola, L., and Britter, R.: Dynamics and dispersion modelling of nanoparticles from road traffic in the urban atmospheric environment – A review, J. Aerosol Sci., 42, 580–605, doi:10.1016/j.jaerosci.2011.06.001, 2011.

Zoumakis, N. M.: A note on average vertical profiles of vehicular pollutant concentrations in urban street canyons, Atmos. Environ., 29(25), 3719–3725, doi:10.1016/1352-2310(95)00105-8, 1995.

---

## Author Comment (AC1)

**Authors' reply to comments from anonymous referees #1 and #2, on the manuscript "Modelling concentration heterogeneities in streets using the street-network model MUNICH"**

The authors thank the referees for the opportunity to submit a revised draft of our manuscript titled "Modelling concentration heterogeneities in streets using the street-network model MUNICH" to Geoscientific Model Development. We appreciate the time and effort that the referees have dedicated to providing valuable feedback on the manuscript. We have incorporated changes to reflect most of the suggestions provided by the referees.

Here is a point-by-point response to the referees' comments and concerns.

**Referee #1**

**General comment**: Distribution of air pollution in street canyon environments are highly heterogeneous. Homogenous assumption may be inappropriate to capture large concentration gradient in the streets, especially in narrower streets. An advanced version of the street-network model MUNICH (Model of Urban Network of Intersecting Canyons and Highways) was developed in this paper to achieve the representation of concentration gradients in streets. The model was then applied to real streets. Simulated results were compared with observations and other models, i.e. Operational Street Pollution Model (OSPM) and homogenous version of MUNICH, showing the advantages of the advanced MUNICH. A sensitivity test with and without street network shows the need to use a street network (rather than a single street) in the MUNICH. There are some comments as follows:

**Comment 1**: Line 70: Apart from the chemical transformation, were the aerosol processes as mentioned in Lines 43-44 also considered for PM in this study?
**Response:** Yes, aerosol processes for PM are considered in this study thanks to the coupling with SSH-aerosol. Sentence line 77 is modified to "The transport term includes advection from one street to another and vertical transport between the street and the background above it. The chemistry includes gas-phase chemistry, as well as aerosol dynamics (coagulation, condensation/evaporation)".

**Comment 2**: Line 117: Why 3 vertical levels? Can it be increased to capture more vertical variation if needed?
**Response:** The number of vertical levels can be increased in MUNICH-hete as the vertical variations within the streets of winds and mixing lengths are parameterized when discretizing the streets. However, the first vertical level at the bottom of the street should not be too thin because of mixing due to traffic turbulence. A minimum height for the first layer of 1.5 m seems reasonable. Furthermore, increasing the number of vertical level would also increase the simulation time.
The following sentences are added at line 139, "Note that more than three vertical levels could be defined in MUNICH-hete, as the vertical variations within the streets of winds and mixing lengths are parameterized when discretizing the streets. However, the first vertical level at the bottom of the street should not be too thin because of mixing due to traffic turbulence. A minimum height for the first layer of 1.5 m seems reasonable."

**Comment 3**: Line 181: Were the simulations conducted in every hour over the year?
**Response:** In all three cases, simulations were conducted every hour. For the Danish cases, simulations cover the whole year 2019, whereas for the Paris case, simulation covers three months. The following sentences are modified at line 195, "Simulations are performed over the year 2019 to generate hourly concentrations for two street networks in Copenhagen, Denmark." and at line 287, "Simulations are performed from 22 March to 15 June 2014 to generate hourly concentrations with [...]"

**Comment 4**: Lines 192-195: please provide readers with a bit more details about WRF, UBM and OML-Highway model.

**Response:** Paragraph line 206 has been modified to "MUNICH and OSPM use the same input data to estimate the street concentrations. The meteorological parameters originate from simulations performed with the Weather Research and Forecasting model (WRF) (Skamarock et al., 2008). The background concentrations are simulated using the Urban Background Model (UBM) (Berkowicz et al., 2000), which is "a multiple source model that applies a Gaussian approach for horizontal dispersion and a linear approach for vertical dispersion up to the boundary layer" (Jensen et al., 2016). Traffic emissions are generated using the procedure implemented in the local-scale Gaussian air pollution model OML-Highway (Olesen et al., 2015) allowing for precise information for each street segment. Traffic data (average daily traffic, travel speed, and share of heavy-duty vehicles) are used to generate emissions by use of the European emission model COPERT IV. Traffic emissions include exhaust emissions and brake, tyre and road wear."

Comment 5: Line 204-206: what are the values for H and W? Please provide more details about how the street parameters are derived for the street network (as indicated in Fig.4)?
**Response:** Dimensions of the segments for which observations are available have been added for all three cases. The following sentences are added line 213, "The street parameters (building height and street width) used in MUNICH originate from the OSPM setups." and line 286, "The street parameters (building height and street width) were obtained from the BD TOPO database (https://geoservices.ign.fr/bdtopo)."

Comment 6: Line 215-216: The underestimation for NO2 may be also attributed to the missing of VOC chemistry.
**Response:** The sentence line 233 has been modified to: "[...] at the regional scale; but also to the absence of volatile organic compounds in the simulation.".

Comment 7: Line 221: OSPM can capture the difference horizonal heterogeneities in the street (as mentioned in Line 212), while MUNICH-hete-l1 can only represent the average in the whole of first vertical layer. How can it be considered as improvement for MUNICH in this aspect?
**Response:** At this stage of development, MUNICH-hete-l1 represents the horizontal average in concentrations for the first layer. This is a first step towards improving on the assumption of vertical and horizontal homogeneity. When compared to MUNICH-homo in regard to observations or OSPM results, MUNICH-hete-l1 displays an improvement of the concentrations.

Comment 8: Line 268: Are $PM_{2.5}$ and $PM_{10}$ considered as passive tracers, or with aerosol processes?
**Response:** For the Paris case, simulations are performed with SSH-aerosol (Sartelet et al. 2020) to represent both the gas-phase chemistry and the aerosol processes. A sentence has been added in line 289: "For this case, MUNICH is coupled to SSH-aerosol Sartelet et al. 2020 to represent gas-phase chemistry and aerosol dynamics.".

Comment 9: Fig. 7: units were not shown clearly.
**Response:** Units have been correctly added to the figure.

Comment 10: Line 314: Please add discussion about the influence of aspect ratio (AR). Which AR is favourable for the MUNICH-hete?
**Response:** A discussion on the impact of the discretization introduced in this work on street concentration, sorted by different classes of aspect ratio, has been added.

Comment 11: Line 331: Also to include VOC chemistry for the next step?
**Response:** For the Danish cases, VOC information, such as background concentrations and emissions in the street, is not available. MUNICH simulations could be performed with VOC concentrations, if background concentrations and emissions are available. VOCs are included for the Paris case. The following sentence is added in line 376, "Finally, chemistry of volatile organic compounds could be added in the Danish cases if the

background concentrations and emissions are available.".

**Referee #2**

General comment: A new version of the street-network model MUNICH is presented that introduces a vertical discretization of street canyons in three layers and the horizontal discretization of the street canyon in two zones (recirculation and ventilation) based on well-evaluated parameterizations of the OSPM model. The manuscript is well organized although more discussion of pertinent issues such as the influence of the vehicle wake and atmospheric stability on the vertical pollutant distribution in street canyons would give the paper more credibility. The new version, termed MUNICH-hete is applied to two street canyons and districts in Copenhagen and to one district in Greater Paris and compared to the homogeneous version of the model and OSPM results for two street-level receptors as well as street-level measurements. Further, a sensitivity study of the influence of the street network on the concentrations in the streets with MUNICH-hete was performed concluding that intersections with neighboring streets underline the need for simulating a street network rather than a single street.

MUNICH-hete predicts that concentrations of compounds emitted by traffic are higher at the bottom of the street than at the top. However, the vertical distribution of pollutants in the three-layer street canyon is not compared to measurements. For wide streets and avenues with dense traffic, the concentrations in the first level (MUNICH-hete-11) are generally higher than in the homogenous version of the model. A systematic analysis of the influence of the aspect ratio on the concentration differences between MUNICH-hete-11 and the homogenous version has unfortunately not been performed.

Comment 1: P2, line 35-36: EPISODE-CityChem (Karl et al. 2019) that uses a simplified OSPM and CALIOPE-Urban (Benavides et al. 2019), which applies a downscaling depending on atmospheric stability and building density, should be mentioned here.
**Response:** Both models (with references) have been added to the article, line 39.

Comment 2: P2, line 42: Was the SSH-aerosol model used in this study? It would be interesting to see the effect of detailed VOC chemistry on SOA concentrations along street canyons.
**Response:** SSH-aerosol has been used for the Paris case. For the two Danish cases, information on VOCs and PM is not available.

Comment 3: P3, line 60: "The new version of MUNICH" should be given a version number. Is it based on MUNICH v2.0 in all other aspects than the vertical discretization or is it a different development branch?
**Response:** This new version of MUNICH is labelled hete. The following was added in the manuscript, line 70, "The heterogeneous version was developed from MUNICH v2.0." The new developed model functionalities will be integrated in the next official MUNICH version.

Comment 4: Resuspension of road dust can be an important of the non-exhaust particulate matter emissions from vehicles and its relevance will increase in the future due to transition to electric mobility. Here it is stated that resuspension is a minor process compared to transport and chemistry, however that only applies to vehicular exhaust emissions. The consequence would be to restrict the application of MUNICH-hete to vehicular exhaust emissions of gaseous pollutants.
**Response:** In previous work (Lugon et al. 2021), resuspension of road dust was explicitly simulated in MUNICH by estimating first the concentrations that deposit on the street and then resuspending the deposited concentrations depending on the traffic flow. The influence on BC concentration was low. Note that non-exhaust emissions due to brake, road and tyre wears are taken into account.

**Comment 5**: How is the influence of atmospheric stability on vertical mixing within a street canyon taken into account in MUNICH? Some dispersion models like SIRANE account for this.

**Response:** In MUNICH-hete, the influence of the atmospheric stability on vertical mixing is taken into account by modifying the standard deviation of the vertical wind velocity at roof level and thus the vertical transfer rate depending on the length of Monin-Obukhov, as in MUNICH-homo. The following sentences are added in line 176, "The influence of the atmospheric stability on vertical mixing is taken into account by modifying the standard deviation of the vertical wind velocity at roof level and thus the vertical transfer rate depending on the length of Monin-Obukhov, as in MUNICH-homo."

**Comment 6**: Vertical distribution of pollutants in the street canyons of this study with MUNICH-hete should be shown, and compared to observation in different heights of the street canyons. There is some debate about the shape of the vertical profile in street canyons. Zoumakis 1995 based on analysis of measurements report that the average vehicular pollutant concentration profile follows the general exponential form rather than the simple exponential function or Gaussian distribution. Kumar et al. 2009 report that concentrations of particle numbers increase from road level up to 2 m height, this increase is reproduced by a CFD model.

**Response:** For all three cases, we do not have observations at different heights. The simulated vertical profile is highly dependent on the shape of the mixing length used in the modelling. The mixing length used here has been parameterized from CFD modelling, as detailed in Maison et al. 2022. Typical vertical profiles are displayed in Maison et al. 2022. However, in the present work, by implementing a vertical discretization based on just three vertical levels, we are not seeking to reproduce finely the vertical profile. The main objective is to improve the representation of concentrations close to the ground by avoiding the excessive dilution associated with the homogeneity assumption. To emphasise this point, we have added the following sentence in the introduction, line 55, "With this discretization we do not aim to reproduce finely the vertical profile of concentrations. The main objective is to improve the representation of concentrations close to the ground by avoiding the excessive dilution associated with the homogeneity assumption."

**Comment 7**: This also concerns the question whether three levels are sufficient to represent the pollutant distribution in a street canyon. As I understand it, the height of the first level defines the volume in which traffic emissions are injected and diluted. The height of the first layer should be evaluated with a model that can resolve the flow and dispersion in the vehicle wake (Kumar et al. 2011). A sensitivity study on the height of the munich-11 level should be carried out.

**Response:** Indeed, the height of the first layer defines the volume into which traffic emissions are instantly diluted. We recognise that the choice of a height of 2 m for the simulations presented is partly arbitrary. However, as mentioned in the reply to a previous comment, and as added in the manuscript at line 139, the first level of MUNICH-hete should not be too fine because of mixing due to traffic turbulence. We indicate than 1.5 m, as a representative height of the vehicles, is probably a good lower limit. We rely in practice on the numerical dilution to get a rough implicit representation of the traffic induced turbulence. If the height of the first layer is too high, traffic emissions are artificially diluted vertically. Therefore, the height of the first level cannot vary much in the framework of the simple Eulerian description we adopted to ensure low computational burdens.

**Comment 8**: How is traffic-induced turbulence considered in the vertical turbulence flux sigma_w of Equation 12? The influence of the traffic-induced turbulence should then only be considered in the first level.

**Response:** The following sentence has been added to line 128, "This traffic-induced turbulence is not explicitly considered in the model." The model was built from CFD simulations (see Maison et al. 2022) by parameterizing the vertical mixing length.

**Comment 9**: P7, line 173: Explain no-slip condition at the ground.

**Response:** The no-slip condition refers to the value of the wind at the ground ($u(z = 0) = 0$). The sentence "the no-slip condition at the ground that is always satisfied" has been replaced by "the no-slip condition at the ground that is always satisfied ($u(z = 0) = 0$)" at line 186.

**Comment 10**: Is it planned to use 3-D building heights to inform the model on the real street canyon geometries?

**Response:** Real street canyon geometries are used. They are defined from real building heights, which are averaged over each street segment to determine the average building height. Sentence line 88 has been modified to "[...] with $H$ the average building height over the street segment [...].".

**Technical comment 1**: P1, line 10: "Results show an improvement". In the abstract, give quantitative information on the improvement and state compared to which model the improvement was achieved.

**Response:** The following sentence has been added line 11 "These increases reach up to $60\,\%$ for $NO_2$ and $30\,\%$ for $PM_{10}$ comparatively to MUNICH v2.0.".

**Technical comment 2**: P9, line 226: "when $NO_2$ concentrations are underestimated" – does this refer to measured NO2?

**Response:** This refers to both measured $NO_2$ and OSPM-simulated $NO_2$. The sentence line 245 has been modified to "[...] when $NO_2$ concentrations are underestimated in MUNICH-hete compared to observations and OSPM concentrations."

**Technical comment 3**: Figure 3: Annotations on x-axis and y-axis are incomplete.

**Response:** Annotations have been correctly added to the figure.

**Technical comment 4**: Figure 5: Same as for Figure 3. In Figure 5a the line for the CO observations is missing.

**Response:** Annotations have been correctly added to the figure. Concentrations of CO were not measured for the Jagtvej case. Text line 261 has been modified to indicate it.

**Technical comment 5**: Figure 7: Same as for Figure 3.

**Response:** Annotations have been correctly added to the figure.

**References**

Benavides, J. et al. (2019). "CALIOPE-Urban v1.0: coupling R-LINE with a mesoscale air quality modelling system for urban air quality forecasts over Barcelona city (Spain)". In: *Geosci. Mod. Dev.* 12.7, pp. 2811–2835. DOI: 10.5194/gmd-12-2811-2019.

Karl, M., S.-E. Walker, S. Solberg, and M. O. P. Ramacher (2019). "The Eulerian urban dispersion model EPISODE – Part 2: Extensions to the source dispersion and photochemistry for EPISODE–CityChem v1.2 and its application to the city of Hamburg". In: *Geosci. Mod. Dev.* 12.8, pp. 3357–3399. DOI: 10.5194/gmd-12-3357-2019.

Kumar, Prashant, Andrew Garmory, Matthias Ketzel, Ruwim Berkowicz, and Rex Britter (2009). "Comparative study of measured and modelled number concentrations of nanoparticles in an urban street canyon". In: *Atmos. Environ.* 43.4, pp. 949–958. DOI: https://doi.org/10.1016/j.atmosenv.2008.10.025.

Kumar, Prashant, Matthias Ketzel, Sotiris Vardoulakis, Liisa Pirjola, and Rex Britter (2011). "Dynamics and dispersion modelling of nanoparticles from road traffic in the urban atmospheric environment—A review". In: *Journal of Aerosol Science* 42.9, pp. 580–603. DOI: https://doi.org/10.1016/j.jaerosci.2011.06.001.

Lugon, L., J. Vigneron, C. Debert, O. Chrétien, and K. Sartelet (2021). "Black carbon modeling in urban areas: investigating the influence of resuspension and non-exhaust emissions in streets using the Street-in-Grid model for inert particles (SinG-inert)". In: *Geosci. Mod. Dev.* 14.11, pp. 7001–7019. DOI: 10.5194/gmd-14-7001-2021.

Maison, Alice, Cédric Flageul, Bertrand Carissimo, Andrée Tuzet, and Karine Sartelet (2022). "Parametrization of Horizontal and Vertical Transfers for the Street-Network Model MUNICH Using the CFD Model Code_Saturne". In: *Atmosphere* 13.4, p. 527. DOI: 10.3390/atmos13040527.

Sartelet, K., F. Couvidat, Z. Wang, C. Flageul, and Y. Kim (2020). "SSH-Aerosol v1.1: A Modular Box Model to Simulate the Evolution of Primary and Secondary Aerosols". In: *Atmosphere* 11.5. DOI: 10.3390/atmos11050525.

Zoumakis, N.M. (1995). "A note on average vertical profiles of vehicular pollutant concentrations in urban street canyons". In: *Atmos. Environ.* 29.24, pp. 3719–3725. DOI: https://doi.org/10.1016/1352-2310(95)00105-8.